# FOLD: FAST CORRECT SPECULATIVE DECODING

## ABSTRACT

Speculative decoding accelerates Large Language Model (LLM) inference by using a small, fast 'draft' model to propose tokens that a larger 'target' model then verifies in a single, parallel step. While this paradigm has become the standard for high-throughput inference, the community's focus has been almost entirely on a single metric: maximizing the acceptance rate of drafted tokens. We argue this is a critical oversight. The true bottleneck is not just acceptance, but the catastrophic computational cost of rejection. A single rejected token triggers a cascading failure, discarding all subsequent work and nullifying potential gains. We introduce **F**ast c**O**rrect specu**L**ative **D**ecoding (**FOLD**), a framework that fundamentally reframes the problem from merely avoiding rejection to instantly recovering from it. FOLD transforms the verification step itself. Instead of a simple pass/fail check, our novel verifier uses an integrated Early Exit module to proactively generate high-probability alternative sequences in parallel. When the primary draft fails, FOLD doesn't discard the computation; it seamlessly pivots to a pre-computed, correct path. This turns a catastrophic failure into a minor course correction, salvaging the entire speculative branch. Extensive experiments show that by treating rejection as an opportunity for correction, not a point of failure, FOLD achieves up to a $4.09\times$ speedup over Auto Regression decoding, setting a new bar for inference efficiency. We anonymously open-source our project at `https://anonymous.4open.science/r/iclr26-fold`.

## 1 INTRODUCTION

Modern large language models (LLMs) (OpenAI et al., 2024) such as DeepSeek-V3 (DeepSeek-AI et al., 2024), Qwen-3 (Yang et al., 2025), and LLaMA-3 (Grattafiori et al., 2024) have demonstrated exceptional performance and are widely applied across various domains. During the Auto Regression generation process, each token is generated by invoking all model parameters, and the length of text produced in a single dialogue can range from hundreds to thousands of tokens. As the model parameters must be repeatedly loaded from memory during decoding, memory bandwidth limits overall inference speed (Leviathan et al., 2023).

Speculative decoding (Leviathan et al., 2023)methods aim to address this issue by innovatively splits LLM inference into two phases—drafting and verification— which rapidly generate fixed $\gamma$ draft tokens and then verify them in parallel (Li et al., 2024b). Thanks to the tiny model size compared to the target model, it brings notable acceleration benefits, while effectively improving GPU utilization and accelerating LLM inference.

Previous speculative decoding efforts have consistently focused on improving the effectiveness of the draft. EAGLE series (Li et al., 2024a) and HASS (Zhang et al., 2025) leverages features from the Target model to design and train more advanced draft models. AdaEAGLE (Zhang et al., 2024) leverages the LDLP module to explicitly predict the optimal number of draft tokens during inference to guide the draft model. Meanwhile, Ouroboros (Zhao et al., 2024) only accelerates draft stages, by caching previous draft tokens. The core idea of these approaches is that the acceleration effect of speculative decoding improves as the acceptance rate of draft tokens increases. However, we observe that as the draft token sequence length grows, the probability of rejection also increases. Moreover, when rejection occurs, the subsequent sequence is discarded. This results in significant GPU computation time being wasted, which negatively impacts inference acceleration. Conversely, if this computation time can be utilized, the acceleration effect can be further enhanced. Although Bach-

mann et al. (2025) proposed using Judge Decoding to avoid rejection of draft tokens that are correct but inconsistent with the target model, this leads to it not being a lossless acceleration method.

Based on this insight, we propose **F**ast c**O**rrect specu**L**ative **D**ecoding (**FOLD**), a method that rapidly corrects erroneous draft tokens through an Early Exit Module and enables the draft model to quickly generate subsequent sequences. Specifically, FOLD extends the traditional speculative decoding process from two stages (draft-verify) to four stages-Early Draft, Early Verify, Draft Correct, Final Verify, achieving: (i) extracting the top $k$ early verification results using the logits of the Early Exit Module in stage Early Verify; (ii) integrating multiple potentially correct branches in stage Draft Correct and parallelly computing the subsequent sequences of multiple potential branches using the tree-based attention mechanism (Cai et al., 2024)(i.e. Tree Attention); (iii) selecting the longest accepted sequence branch as the correct branch during Final Verify. This approach ensures that even when the original draft tokens are rejected, while the Early Exit Module generates correct results, the subsequent draft token sequences do not need to be discarded, as their prefix is correct. Furthermore, FOLD is compatible with most existing speculative decoding methods. We choose Pearl(Liu et al., 2025) as an example for adaptation and test it on various text generation datasets, achieving outstanding acceleration, with up to a $4.09\times$ improvement in inference speed.

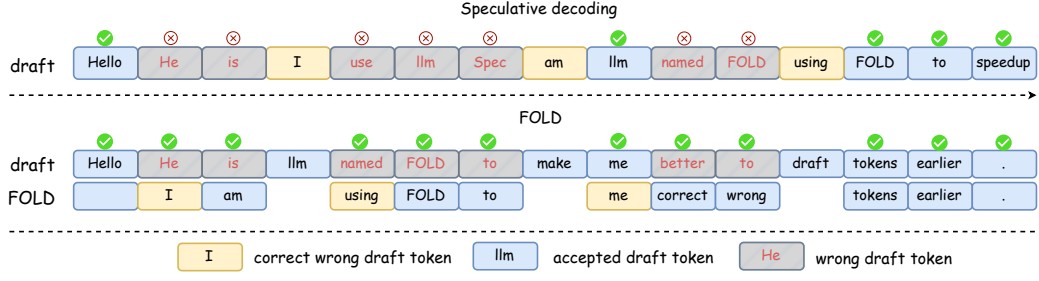

Figure 1: An overview of speculative decoding(SD) and our FOLD. SD corrects wrong draft tokens after per draft-verify turn finish, while FOLD instantly corrects wrong draft tokens during draft phase.

## 2 BACKGROUND

**Notations.** In this paper, we use $\gamma_1$ and $\gamma_2$ to represent the number of forward passes performed by the draft model during the Early Draft and Draft Correct stages respectively, while $\gamma$ denote farward times of draft model per round in Speculative Decoding.

**Speculative decoding.** Speculative decoding is an innovative technique for enhancing the efficiency of auto-regressive large language models (LLMs) without sacrificing output quality. This approach utilizes a smaller, efficient "draft model" to predict multiple subsequent tokens, which are then validated in parallel by the target LLM. By doing so, speculative decoding enables the generation of multiple tokens within the time typically required for a single inference. Formally, the speculative decoding framework consists of two key stages: draft construction and draft verification (Xu & McAuley, 2023b).

**Early Exit techniques.** Early Exit techniques use an auxiliary module to stop an LLM's inference at an intermediate layer for simpler inputs, avoiding unnecessary computation. This balances the power of large models, needed for complex tasks, with the efficiency of shallower networks for "easy" examples (Xu & McAuley, 2023a). Bae et al. (2023) proposes a fast and robust early-exiting framework for autoregressive language models, using synchronized parallel decoding to improve inference speed and efficiency while maintaining accuracy. Liu et al. (2024b) proposes a speculative decoding method for faster LLM inference using early exiting and a Thompson sampling control mechanism to balance speed and accuracy.

**Pearl** PEARL (Liu et al., 2025) introduces a pre-verification strategy to validate the initial draft token during the drafting phase and a post-verification approach to generate additional draft tokens

during the verification phase. By implementing these strategies, PEARL enables parallel execution of the drafting and verification phases, while adaptively adjusting the draft length to suit different scenarios. This effectively mitigates the issue of mutual waiting between phases.

## 3 FOLD

We delineate the core architecture of FOLD in Section 3.1, followed by demonstrating its seamless integration with speculative sampling methods through the Pearl case study in Section 4, where we also furnish theoretical proofs validating its efficacy.

### 3.1 ARCHITECTURE

FOLD splits the conventional draft-verify pipeline into four distinct unit: *Early Draft*, *Early Verify*, *Draft Correct*, and *Final Verify*, while stage combination of *Early Verify*, *Draft Correct* is so flexible that we will present in the following section 4 to show how FOLD Combined with Pearl. By decomposing both drafting and verification into alternating unit, this architecture explicitly mitigates redundant computation in draft models while reducing unnecessary target model verifications. The FOLD framework, compared to traditional speculative sampling methods, decomposes the draft stage into Early Draft and Draft Correct. Correspondingly, in traditional speculative sampling methods, the $\gamma$ tokens generated by the draft model in each draft stage are also split into generating $\gamma_1$ and $\gamma_2$ draft tokens, , which means draft model forward times during Early Draft and Draft Correct phase respectively, i.e. $\gamma = \gamma_1 + \gamma_2$.

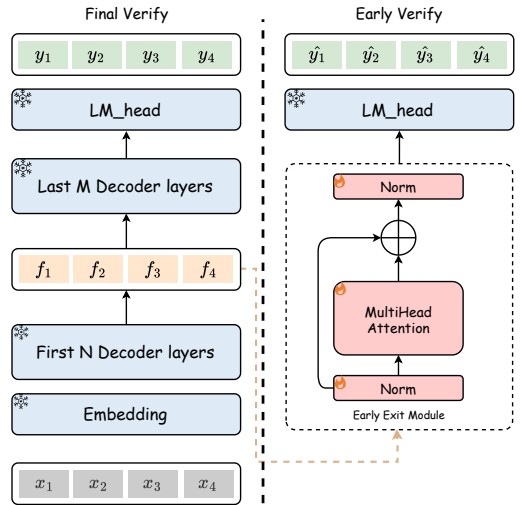

Figure 2: Architecture of FOLD. The Early Exit Module uses the hidden states from the first N layers as input, where "snow" denotes parameter freezing and "fire" denotes trainable parameters.

**Early Draft.**   As with conventional speculative sampling methods, Early Draft phase utilizes the draft model to generate standard draft token sequences, while executing fewer inference steps $\gamma_1$ in this drafting stage compared to traditional speculative sampling approaches.

**Early Verify.**   For unverified tokens, Early Verify performs validation during this phase. The Early Exit Module executes only once per verification round to produce preliminary validation features, while saving First N layers hidden state for Final Verify phase as show in Figure 2. The features of early exit module are subsequently fed into the target model's lm_head to generate early exit logits. We extract the top $k$ most probable tokens from these logits as candidate predictions, serving as backup verification options during *Final Verify* procedures, effectively mitigating performance degradation in cases of draft token rejection.

To simplify this work, we adopt Kangaroo's (Liu et al., 2024a) adapter module as our early exit module. Specifically, for target models, our observations indicate that employing excessively shallow decoder layers results in degraded token quality and impaired alignment with the target model. Therefore, selecting the first $N$ layers close to half the total layers (i.e., $\frac{\text{total layers}}{2}$) is crucial to strike an optimal balance between early exit token quality and the number of subsequent speculative tokens for next verification turn.

**Draft Correct.**   As shown in 3, during the Draft Correct phase, the draft model synthesizes draft tokens from Early Verify with early exit tokens to generate subsequent speculative tokens. Integrating these candidate tokens with the draft model's native output sequence, our framework constructs

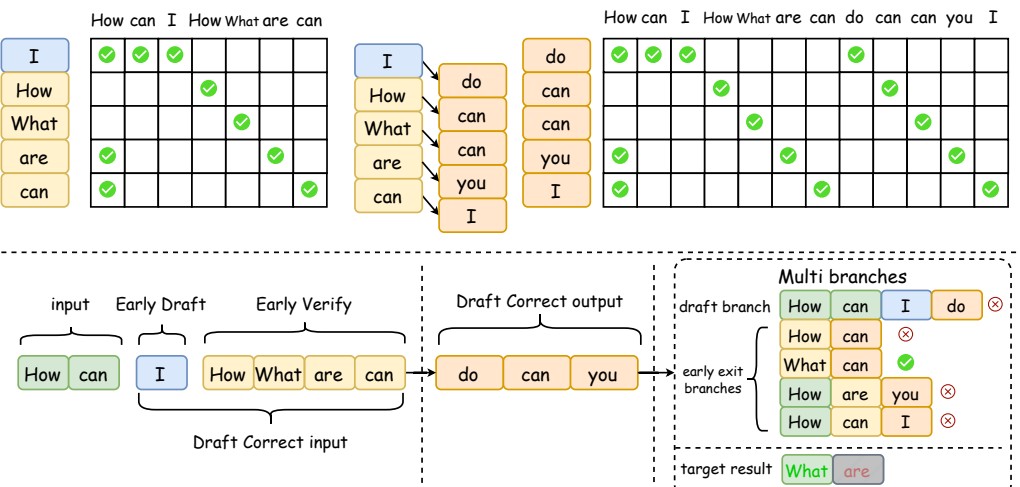

Figure 3: Diagram of the principle during Draft Correct phase. In the Draft Correct stage, the input is integrated from multiple branches, with attention causal masks designed based on the branches, resulting in the output of subsequent tokens for the corresponding branches. Green squares represent the current input round, blue squares represent the Early Draft, yellow squares represent the Early Verify, and orange squares represent the Draft Correct outputs.

$\hat{\gamma} \times K + 1$ draft branches, where $\hat{\gamma}$ denotes the token length verified by Early Verify. Utilizing tree-based attention mechanism, the draft model processes all branches concurrently, predicting $\gamma_2$ subsequent tokens per branch for Early Verify in next round.

Specifically, within the FOLD framework, $\gamma_1$ and $\gamma_2$ — representing the forward times of draft models during the Early Draft and Draft Correct phases respectively — are both assigned a value of 1, and the tokens for current verification inputs are "How" and "can". The early exit layer selects the top $k = 2$ candidates from logits— specifically, candidate pairs of ("How", "What") and ("are", "can"). Simultaneously, the draft model outputs the token "I" during the Preliminary Verification phase. These are subsequently integrated to form $\gamma \cdot k + 1 = 5$ distinct branches with initial tokens: "I", "How", "What", "are", and "can". In this phase, the draft model employs tree-based attention mechanism to infer sequences $\gamma_2 = 1$ time steps, yielding subsequent tokens for all branches such as '"do can can you I"'. Later, upon the Target model computing the full verification results of "What" and "are", the token "How" of draft branch is rejected. Fortunately, Early Verify obtained the correct answer and is thus able to immediately correct the erroneous draft token, while also obtaining subsequent draft tokens "can" during the Draft Correct phase.

**Final Verify.** During Final Verify phase, target model resumes computation from the Early Verify's first N layers hidden states bypassing the early-exit module, ultimately selecting the longest draft branch as correct branch.

In summary, this architecture selects a top $k$ candidate subset from the early exit module's logits in Early Verify phase. During the Draft Correct phase, it employs a tree-based attention mechanism to simultaneously predict multiple branches—including the draft model's original inference path and correction paths derived from the early exit module. This approach preserves computational efficiency when rejection occurs by dynamically recovering from rejected tokens via alternative branches. In contrast, traditional speculative decoding methods typically discard tokens upon rejection.

## 4 FOLD WITH PEARL

As a representative work in this domain, we make Pearl as our base method of choice in this chapter to demonstrate FOLD's adaptability within speculative sampling frameworks.

## 4.1 PEARL

Pearl designs a strategic combination of pre-verify and post-verify, achieving the parallelization of draft-verify.

**Pre-verify.** When downgrade to pre-verify with rejection occuring, the degraded performance where target model verifies merely next one token per forward pass, which ultimately bottlenecks Pearl's achievable speedup.

**Post-verify.** In optimal scenarios with all draft tokens accepted in previous verification, Pearl achieves maximum acceleration by enabling the target model to verify $\gamma$ tokens per forward pass during continuous post-verification operations.

By introducing parallelization, it supports fine-grained progressive validation of $x$ draft tokens ($x \in \{1, \gamma\}$).

$$x = \begin{cases} 1 & \texttt{mode} = \texttt{pre-verify} \\ \gamma & \texttt{mode} = \texttt{post-verify} \end{cases}$$

If any token fails verification, the entire subsequent sequence undergoes invalidation with mode downgrading to pre-verify for only next token verification.

## 4.2 FOLD FIT IN PEARL

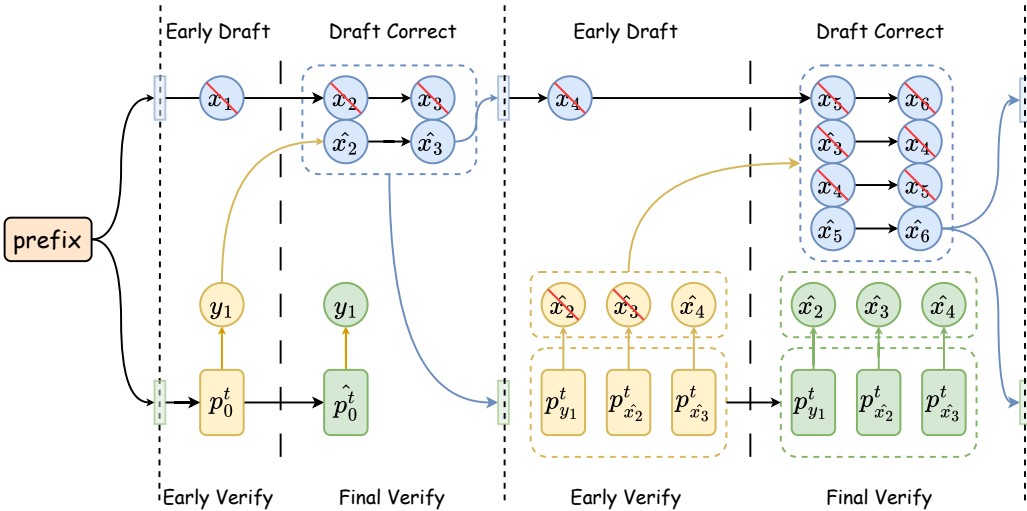

Figure 4: FOLD with Pearl

As depicted in Figure 4, we extend Pearl's original two-stage draft-verify paradigm to a four-stage framework: Early Draft, Early Verify, Draft Correct, and Final Verify. With FOLD architecture, Pearl's drafting phase—which originally generated $\gamma$ tokens—is refactored such that the Early Draft and Draft Correct modules yield $\gamma_1$ and $\gamma_2$ tokens, respectively. During the Draft Correct stage, the model merges multiple branches from Early Draft and Early Verify, and selects the branch with the longest accepted sequence to advance to the next round of verification based on validation in the Final Verify stage.

In cases where tokens from the draft branch do not pass verification, yet the early exit branch from rapid correction contains the correct token output, the next verification cycle acts only on the tokens generated during Draft Correct phase, thereby bypassing excess pre-verify owing to rejection. Importantly, rejection occurs solely when the branch with the longest accepted sequence corresponds to the original draft branch and the acceptance length falls short of $\gamma$.

### 4.3 ANALYZE OF FOLD WITH PEARL

#### 4.3.1 IMPACT OF REJECTION FOR PEARL

To quantify the impact of the pre-verify strategy for Pearl on overall inference performance, we conduct the following analysis: For a given prefix, the next token generated by the LLM are deterministic with greedy sample strategy, resulting in a fixed number and pattern of rejections during the draft-verify process. Furthermore, due to Pearl's fine-grained verification of draft tokens, we establish two premises:

1. The number of times Pearl enters the pre-verify phase due to rejected draft tokens is constant, i.e.$C_{\text{pre}}$ is constant.

2. For a fixed-length response, the relationship between the number of tokens undergoing verification ($N_{\text{verified}}$) and the number of tokens accepted by the target model ($N_{\text{accepted}}$) satisfies: $N_{\text{verified}} = N_{\text{accepted}} + \lambda$, where $\lambda$ is a constant.

Concurrently, we derive following fundamental equations:

$$C_{\text{total}} = C_{\text{post}} + C_{\text{pre}} \tag{1}$$

$$N_{\text{verified}} = \gamma \cdot C_{\text{post}} + C_{\text{pre}} \tag{2}$$

$$N_{\text{accepted}} = N_{\text{verified}} + \lambda \tag{3}$$

where $\gamma$ represents draft tokens generated by draft model during per target model forward; $C_{\text{total}}$ represents total target model forward times; $C_{\text{post}}$ represents post-verify operations which verify $\gamma$ tokens each; $C_{\text{pre}}$ represents pre-verify operations which verify 1 token each; $N_{\text{verified}}$ represents total tokens involved in verification; $N_{\text{accepted}}$ represents total tokens accepted in verification. Both $N_{\text{verified}}$ and $N_{\text{accepted}}$ remain constant.

Solving the above system yields the fundamental performance equation:

$$C_{\text{total}} = \frac{N_{\text{verified}}}{\gamma} + \frac{\gamma - 1}{\gamma} \cdot C_{\text{pre}} \tag{4}$$

Equation equation 4 characterizes the correlation between the count of rejection $C_{\text{total}}$ and the count of pre-verify $C_{\text{pre}}$, effectively quantifying the verification burden imposed on the target model. Crucially, its number of rejection serves as the primary determinant of PEARL's acceleration performance.

#### 4.3.2 INCREASE OF FOLD

For comparison, with the addition of FOLD, we can derive the following fundamental equations:

$$\gamma = \gamma_1 + \gamma_2 \tag{5}$$

$$C_{\text{pre}} = C_{\text{early}} + \hat{C}_{\text{pre}} \tag{6}$$

$$C_{\text{total}} = C_{\text{post}} + C_{\text{early}} + \hat{C}_{\text{pre}} \tag{7}$$

$$N_{\text{verified}} = \gamma \cdot C_{\text{post}} + \gamma_2 \cdot C_{\text{early}} + \hat{C}_{\text{pre}} \tag{8}$$

Solving this system yields the performance equations:

$$C_{\text{post}} = \frac{N_{\text{verified}} + (1 - \gamma_2)C_{\text{early}} - C_{\text{pre}}}{\gamma} \tag{9}$$

$$C_{\text{total}} = \frac{N_{\text{verified}} + (1 - \gamma_2)C_{\text{early}} - C_{\text{pre}}}{\gamma} + C_{\text{pre}} \tag{10}$$

$$C_{total} \propto (\frac{1 - \gamma_2}{\gamma} \times C_{early}) \tag{11}$$

where $\gamma_1$ represents draft tokens generated in *Early Draft* phase; $\gamma_2$ represents draft tokens generated in *Draft Correct* phase; $C_{\text{early}}$ represents Early Verify operations preventing pre-verify; $\hat{C}_{\text{pre}}$

represents actual pre-verify operations executed; and other symbols maintain consistent definitions from the analyze of the Pearl.

From Equation 10, we can observe that the number of model inferences has an inverse relationship with both $C_{\text{early}}$ and $\gamma_2$ ($\gamma_2 > 1$) as shown in Equation 11. This means that as the success rate of Early Verify increases, the total number of inferences will be effectively reduced, ultimately decreasing the overall inference time.

## 5 EXPERIMENTS

Unlike other speculative decoding methods, FOLD achieves further performance improvements by rapidly correcting erroneous draft tokens, and it is designed to be used in combination with other speculative decoding methods. As a result, its acceleration performance depends on the underlying method and the model pair being used. For example, when adapting with Pearl the combination of Llama3.1-70B and Llama3.2-1B demonstrates remarkable acceleration, even slightly outperforming EAGLE2, whereas the model pair of Llama2-70B and Llama2-7B exhibits more modest acceleration. Therefore, FOLD is only suitable for comparison with other training-free methods, instead of approaches such as EAGLE that focus on training high-quality draft models – though we will still provide EAGLE2's performance metrics as reference.

**Details of target model.** For both the draft and target models, we keep all their parameters frozen and train only the Early Exit Module, as illustrated in Figure 2. To simplify our implementation, we adopt KANGAROO (Liu et al., 2024a) as our Early Exit module.

**Early Exit Module.** We train the Early Exit Module with the AdamW optimizer on the ShareGPT dataset following Medusa, while learning rate set at 5e-5. Considering the scenario of Pearl with FOLD, both the Early Draft & Early Verify steps and the Draft Correct & Final Verify steps require a round of data synchronization. Therefore, it is essential for the Early Exit Module to determine the suitable execution timing based on an appropriate speed ratio. In light of this consideration, and to balance the performance of the Early Exit Module with the number of tokens generated during the Draft Correct stage, we choose to execute the Early Exit Module at the stage corresponding to $\frac{\text{total\_layers}}{2}$. Specifically, executing the Early Exit Module too early allows the draft model to correct a greater number of invalid tokens during the Draft Correct stage, but the quality of Early Exit tokens becomes difficult to guarantee. Conversely, Higher-quality Early Exit tokens may result in the Draft Correct stage producing too few tokens, which ultimately might compromise the overall acceleration effect.

**Models** Due to implementation constraints of the Kangaroo's open-source code, we only conduct experiments using the Llama series models. We use Llama2-70B (Touvron et al., 2023) and Llama3.1-70B as the target models, while Llama2-7B, Llama3.2-1B, and Llama3.1-8B serve as the draft models.

**Baselines** We adopt vanilla Auto Regression decoding as the baseline, establishing it as the reference for speedup ratios (1.00x). We select several training-free speculative decoding baselines for comparison: **Speculative Decoding (Leviathan et al., 2023)**:standalone SD methods, **Ouroboros**[1] (Zhao et al., 2024), **Assisted generation** (Gante, 2023), and Pearl(Liu et al., 2025). Additionally, we report the acceleration performance of **EAGLE2** (Li et al., 2024b) with Llama3-70B just as a reference.

**Metrics.** FOLD does not modify the target model's weights and uses strict speculative sampling acceptance conditions, ensuring no loss in performance. Therefore, we do not evaluate generation quality. Instead, we use the *Speedup Ratio* to assess the acceleration performance, where the speedup ratio is defined as the actual test speedup ratio relative to vanilla Auto Regression decoding. Notably, Pearl achieves parallelization of the draft-verify process; consequently, the concept of average acceptance length, which is measured in traditional speculative decoding experiments as the average

---

[1]Ouroboros implementation requires transformers version of 4.36.2, while Llama 3.1 requires transformers $\geq 4.43.0$

number of draft tokens accepted per verification, no longer accurately reflects the acceleration effect. Therefore, the same as Pearl, we also refrain from measuring the average acceptance length.

**Why is acceptance rate not included?** With the introduction of Early Verify and Draft Correct, the number of draft tokens verified by the target model varies dynamically in each round. Additionally, as the draft and verify phases are executed in parallel, the average accepted length becomes difficult to calculate and does not hold much reference value.

Table 1: Experiment results on GSM8K and HumanEval. We bold the best results for each model combination. Ouroboros is reproduced in their official implementation with default parameters. Assisted Generation and Speculative Decoding are reproduced in Pearl. We also list the result of EAGLE with Llama2-70B and Llama3-70B as reference.

| Model | Method | GSM8K | | HumanEval | |
|---|---|---|---|---|---|
| | | speed(token/s) | $\tau$ | speed(token/s) | $\tau$ |
| Llama2-7-70B | Auto Regression | 13.77 | 1.00$\times$ | 13.77 | 1.00$\times$ |
| | Speculative Decoding | 24.62 | 1.79$\times$ | 23.52 | 1.71$\times$ |
| | Ouroboros | 29.25 | 2.12$\times$ | 39.10 | 2.84$\times$ |
| | Assisted Generation | 24.39 | 1.77$\times$ | 25.27 | 1.83$\times$ |
| | Pearl | 34.94 | 2.54$\times$ | 40.92 | 2.97$\times$ |
| | **FOLD**(Pearl) | **38.80** | **2.82**$\times$ | **43.22** | **3.14**$\times$ |
| Llama3-8-70B | Auto Regression | 14.85 | 1.00$\times$ | 14.85 | 1.00$\times$ |
| | Speculative Decoding | 31.09 | 2.09$\times$ | 31.63 | 2.13$\times$ |
| | Assisted Generation | 29.11 | 1.96$\times$ | 30.71 | 2.06$\times$ |
| | Pearl | 42.65 | 2.87$\times$ | 42.38 | 2.85$\times$ |
| | **FOLD**(Pearl) | **43.43** | **2.92**$\times$ | **45.36** | **3.05**$\times$ |
| Llama3-1-70B | Auto Regression | 14.85 | 1.00$\times$ | 14.85 | 1.00$\times$ |
| | Speculative Decoding | 37.73 | 2.54$\times$ | 46.47 | 3.13$\times$ |
| | Assisted Generation | 37.44 | 2.52$\times$ | 45.80 | 3.08$\times$ |
| | Pearl | 49.04 | 3.30$\times$ | 55.35 | 3.73$\times$ |
| | **FOLD**(Pearl) | **53.49** | **3.60**$\times$ | **60.72** | **4.09**$\times$ |
| Llama2-70B | EAGLE2 | 47.37 | 3.44$\times$ | 53.98 | 3.92$\times$ |
| Llama3-70B | EAGLE2 | 36.53 | 2.46$\times$ | 46.48 | 3.13$\times$ |

Table 2: Speed(token/s) results on MT-bench. We bold the best results for each model combination.

| Model | Method | Writing | Roleplay | Reasoning | Math | Coding | Extraction | STEM | Humanities | Average |
|---|---|---|---|---|---|---|---|---|---|---|
| Llama2-7-70B | SD | 25.95 | 24.74 | 28.98 | 33.83 | 35.18 | 34.84 | 26.73 | 25.95 | 29.51 |
| | Pearl | 32.65 | 31.56 | 36.61 | 43.12 | **43.64** | **43.54** | 33.72 | 33.28 | 37.27 |
| | FOLD | **32.96** | **31.79** | **39.27** | **44.24** | 37.61 | 30.78 | **42.39** | **41.37** | **37.55** |
| Llama3-8-70B | SD | 25.48 | 24.32 | 24.69 | 28.95 | 30.37 | 27.62 | 26.96 | 25.73 | 26.72 |
| | Pearl | 35.08 | 32.68 | 37.29 | 46.73 | **48.77** | **46.69** | 34.97 | 34.39 | 39.59 |
| | FOLD | **39.97** | **36.39** | **46.55** | **52.29** | 43.11 | 32.67 | **51.73** | **51.77** | **44.32** |
| Llama3-1-70B | SD | 28.92 | 26.65 | 32.01 | 41.47 | 44.64 | 40.99 | 28.67 | 27.56 | 33.83 |
| | Pearl | 36.03 | 32.00 | 38.08 | **52.52** | **53.79** | **50.19** | 34.35 | 34.14 | 41.42 |
| | FOLD | **40.15** | **32.47** | **45.50** | 52.10 | 42.11 | 30.92 | **50.70** | **50.65** | **43.09** |

## 5.1 MAIN RESULT

We performed extensive experiments on the aforementioned benchmark tests with NVIDIA H100 GPUs. As shown in Table 1, across various model combinations such as Llama3-1B&70B, Llama3-8B-70B, and Llama2-7B-70B on the GSM8K and HumanEval datasets, FOLD consistently outperformed Speculative Decoding and Pearl in all configurations, achieving a maximum speedup of 4.09 times compared to standard Auto Regression methods and baseline speculative decoding.

In Table 2, using the same model combinations on the MT-bench dataset to test Speculative Decoding, Pearl, and FOLD, FOLD generally achieved superior performance. However, in some categories, FOLD showed only marginal improvements or even slight declines compared to Pearl, which we attribute to the following reasons: (i) certain model pairs, like Llama3 8B&70B, exhibit high consistency in answering simple questions, resulting in fewer opportunities for the Early Exit Module to activate; (ii) integrating FOLD with Pearl adds an extra round of inter-process data synchronization during the Early Draft and Early Verify stages compared to Pearl alone, necessitating

mutual waiting between the draft and target models for completion of the current round or decoder layer, thereby amplifying synchronization overhead; (iii) given the simplicity of the early exit layer structure and limited training data, its accuracy in certain domains is inferior to that of draft models trained on larger-scale datasets with more parameters.

Despite these challenges, FOLD demonstrated excellent overall acceleration performance, showcasing significant potential and superiority for further development, and verifying the feasibility and necessity of rapidly correcting erroneous draft tokens.

## 5.2 ABLATION STUDY

### 5.2.1 FIRST N LAYERS OF TARGET MODEL

Intuitively, using a higher top $k$ parameter in Early Verify to gain more early exit tokens from logits can increase the probability of detecting and correcting erroneous draft tokens. However, excessively high top $k$ values may result in increased inter-process data transmission latency and slow down the inference speed of the draft model. Therefore, we took the Llama2-70B model and its corresponding Early Exit Module as an example to compare the performance impact of Early Verify with different top $k$ parameters. Meanwhile, we control each experiment to output exactly 512 tokens while recording the inference count of the draft model. The results are presented in the table 3.

Table 3: Ablation results of FOLD on GSM8K and HumanEval datasets.

| model | $k$ | GSM8K | | | HumanEval | | |
|---|---|---|---|---|---|---|---|
| | | speed(token/s) | $\tau$ | $M_d$ count | speed(token/s) | $\tau$ | $M_d$ count |
| Llama2-7-70B | 1 | 37.10 | 2.96$\times$ | 69444 | 41.04 | 2.98$\times$ | 79884 |
| | 4 | **38.80** | **2.82$\times$** | 67124 | 43.55 | 3.09$\times$ | 76728 |
| | 6 | 37.84 | 2.75$\times$ | **66740** | **42.79** | **3.10$\times$** | **75988** |

From Table 3, it can be observed that parameter $k$ indirectly improves the overall inference speed by enhancing the success rate of Early Verify corrections. Specifically, disparities in acceleration effects caused by different $k$ values and the draft model $M_d$ count reveal that as parameter $k$ increases appropriately, the number of inferences performed by the draft model gradually decreases. Furthermore, by comparing the performance differences on the HumanEval and GSM8K datasets, we conclude that when the precision of the Early Exit Module is suboptimal in certain scenarios, increasing $k$ to improve the Early Verify accuracy provides greater speed-up benefits than the performance degradation resulting from expanding the input size for single-step inference in the draft model. Thus, it may be necessary to further increase parameter $k$ to observe performance degradation. However, whether an excessively large $k$ holds practical significance remains debatable, especially compared to further improving the accuracy of the Early Exit Module.

# 6 CONCLUSION AND FUTURE WORK

In this paper, we propose FOLD to rapidly correct erroneous draft tokens, using Pearl as an example for adaptation, demonstrating the superiority of this method and its underlying concept. Although experiments revealed that the Early Exit Module exhibits suboptimal accuracy in certain scenarios, thereby affecting performance, this does not detract from the validity of the overall approach. In the future, we will explore how to enhance acceleration effects by improving the Early Exit Module as important part of Early Verify phase. Hope that the ideas represented by FOLD can further enhance the inference speed of large models.

We properly reference all prior methods and datasets employed in our study, using exclusively publicly available data without any utilization of private information. Moreover, we carefully maintain our developed inference acceleration techniques, ensuring their implementation remains free from any discriminatory effects.

## ACKNOWLEDGMENTS

The authors would like to thank all the anonymous reviewers for their insightful comments.

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

## A    ETHICS STATEMENT

This work adheres to the ICLR Code of Ethics. In this study, no human subjects or animal experimentation was involved. All datasets used, including GSM8K, HumanEval and MT-bench, were sourced in compliance with relevant usage guidelines, ensuring no violation of privacy. We have taken care to avoid any biases or discriminatory outcomes in our research process. No personally identifiable information was used, and no experiments were conducted that could raise privacy or security concerns. We are committed to maintaining transparency and integrity throughout the research process.

## B    REPRODUCIBILITY STATEMENT

We have made every effort to ensure that the results presented in this paper are reproducible. All code and datasets have been made publicly available in an anonymous repository to facilitate replication and verification. The experimental setup, including training steps, model configurations, and hardware details, is described in detail in the paper. We have also provided a full description of FOLD, to assist others in reproducing our experiments.

Additionally, the public availability of resources—such as the GSM8K, HumanEval, and MT-bench datasets and early exit modules like Kangaroo—enables consistent and reproducible evaluation results.

We believe these measures will enable other researchers to reproduce our work and further advance the field.

## C    LLM USAGE

Large Language Models (LLMs) were used to aid in the writing and polishing of the manuscript. Specifically, we used an LLM to assist in refining the language, improving readability, and ensuring clarity in various sections of the paper. The model helped with tasks such as sentence rephrasing, grammar checking, and enhancing the overall flow of the text.

It is important to note that the LLM was not involved in the ideation, research methodology, or experimental design. All research concepts, ideas, and analyses were developed and conducted by the authors. The contributions of the LLM were solely focused on improving the linguistic quality of the paper, with no involvement in the scientific content or data analysis.

The authors take full responsibility for the content of the manuscript, including any text generated or polished by the LLM. We have ensured that the LLM-generated text adheres to ethical guidelines and does not contribute to plagiarism or scientific misconduct.

