# OpenReview forum: "FOLD: Fast Correct Speculative Decoding"
_ICLR.cc/2026/Conference — ICLR 2026 Conference Withdrawn Submission_

### Official Review · Reviewer_RFZF · 2025-10-29

**Soundness:** 2
**Presentation:** 1
**Contribution:** 2
**Rating:** 2
**Confidence:** 4

**Summary:**

FOLD is a parallel speculative decoding framework. At the start of a given step, there are possible unverified tokens left over from a previous step. The draft model continues generation from these unverified tokens, while the target model with an early stopping module generates alternatives to the unverified tokens. Next, the full target model continues with verification of the unverified tokens while the draft model continues generation from its own branch (building on top of the unverified tokens) and branches from the early stopped target model (which are alternatives to the unverified tokens). When the full target model finishes verification, the branch with the most fully verified tokens is selected for the next round. Note that that branch will have some tokens not yet verified by the full model (i.e., those generated by the draft model in this round). The authors build FOLD into the PEARL speculative decoding framework with Kangaroo’s early stopping module, and show speedups.

I had difficulty understanding the presentation of the method, so the above is a best-effort summary of the work. If I have misinterpreted the work, please let me know.

**Strengths:**

- The core idea is an effective combination of Kangaroo’s method of early stopping as a draft model, PEARL’s method of parallelizing drafting and verification, and tree-based speculative decoding methods maintaining multiple drafts at once.
- Results are strong, showing reasonably consistent speedups compared to baselines across a variety of tasks.

**Weaknesses:**

The main weakness of the paper is its poor presentation, which makes it difficult to understand and evaluate. For example:
- Lines 58-64 in the intro (“Specifically, FOLD extends … during Final Verify”) do little to explain what the four stages do and discuss concepts before they are introduced (e.g., what constitutes a “verification result”, where do branches come from, how are branches accepted).
- Figure 1 was confusing and did not help me understand the method. The legend was confusing (What’s a “correct wrong draft token”?), and I could not understand where each token came from, nor what the target tokens were. The caption was insufficient, and I did not find any references to the figure in the paper.
- The background was also insufficient. More discussion was needed of related works in Speculative decoding (e.g., the baselines presented in the results). More discussion of Pearl was also needed as it forms one of the core experiments of the paper (though that discussion could have been moved down to Section 4).
- Section 3.1, which describes how FOLD works, was very difficult for me to understand.
  - First, I think discussing the combination of Early Verify and Draft Correct when applying FOLD to Pearl (lines 126-129) this early only serves to confuse the reader.
  - The description of each stage is also unclear. The inputs, models used, methods applied, and outputs from each stage are not at all clear. For example, during Early Verify, it is not clear what tokens are being early verified. On my first read I assumed it was the output tokens from Early Draft, now I think it is unverified tokens from a previous step, but I am still not sure.
  - Figure 3 did not help either. On first read it is unclear how each token is obtained, how branches were built, why the target model re-generated input tokens, etc.
- Similar issues persisted into Section 4. In particular, there was essentially no explanation or discussion for Figure 4, which is a complicated figure key to understanding how FOLD was integrated into PEARL.
- More minor formatting issues:
  - The notations paragraph in the Background section is way earlier than it needs to be and confusingly refers to Early Draft and Draft Correct before they are introduced. Consider integrating it into section 3.1.
  - Also in that paragraph, “farward” should be “forward”.
  - Close quotes are used when open quotes should be used
  - Xu & McAuley, 2023a (line 97) and Xu & McAuley, 2023b (line 101) refer to the same paper, but are cited differently
  - In Line 160, refer to “Figure 3”, not just “3”.
  - Double comma in line 139 (“…draft tokens , , which means…”).
These are just some of the presentation issues I had with the paper. I do not believe the paper is ready for publication in its current state. Significant editing and revising would be necessary to change my opinion.

And some comments on the results section:
- The authors characterize FOLD as a “training-free method” (line 339), but FOLD requires the training of the Early Exit Module. There is a training cost argument to make here the Early Exit Module may be much cheaper to train than a draft model, but training is still required.
- Authors should cite the benchmarks used.
- Experiments are restricted to Llama 3 and Llama 2. Results on models from other model providers would strengthen the results by demonstrating generalizability.
- No confidence intervals are provided for the speedup figures, making it difficult to contextualize the significance of the speedups.

**Questions:**

- Could you clarify the FOLD architecture in detail, specifying the inputs and outputs to each model at each stage, and across steps?
- What is the exact evaluation setup you used? Did all methods have access to the same computational resources?
- Related to this, it seems that FOLD would require more resources at any given time as both the draft model and target model need to run in parallel. Does FOLD actually end up using more GPU hours or does the faster generation offset this additional cost?

---

> ### Author Response · Authors · 2025-11-21
> **Response to Reviewer RFZF with Question1(part 1/3)**
>
> ### 1. Clarification of the FOLD architecture, specifying the inputs and outputs at each stage and across steps.
>
> Here are the figure that show the architecture [https://postimg.cc/w3RTnpVV](https://postimg.cc/w3RTnpVV) and pseudo code [https://i.postimg.cc/7PMcSdqn/FOLD-jia-gou-wei-dai-ma.png](https://i.postimg.cc/7PMcSdqn/FOLD-jia-gou-wei-dai-ma.png). For pseudo code, we will incorporate it into the paper in the future.
>
> FOLD consists of four main stages: **Early Draft**, **Early Verify**, **Draft Correct**, and **Final Verify**. I will detail the inputs, outputs, and roles of each stage below.
>
> Let's assume we have an initial sequence to be verified, $T_0 = \{t_1, t_2, t_3, t_4\}$. We'll also assume the time required for a single inference step is in a 4:1 ratio for the target model to the draft model (i.e., the draft model can complete four inference steps in the time the target model completes one). Further, the time taken for the Early Exit module to produce an output is twice that of a single draft model inference step, as the early exit mechanism utilizes the hidden state from half of the target model's layers.
>
> The workflow begins as follows:
>
> 1.  **Stage 1: Early Draft & Early Verify (Parallel Execution)**
>     *   $T_0$ serves as the concurrent input for both the Early Draft and Early Verify stages.
>     *   **Early Draft**: The draft model takes $T_0$ and autoregressively generates a draft token sequence $T_1 = \{t_5, t_6\}$. This takes two draft model time steps.
>     *   **Early Verify**: The target model's Early Exit module processes $T_0$ and produces two outputs:
>         *   **Cached Hidden States (F)**: A set of hidden states from the initial N layers of the target model, captured *before* the Early Exit block. $F = \{f_1, f_2, f_3, f_4\}$ will serve as the input for the Final Verify stage.
>         *   **Preliminary Logits (L)**: A set of logits, $L = \{l_1, l_2, l_3, l_4\}$, generated by the Early Exit module.
>     *   At this point, the two parallel processes synchronize, awaiting each other's completion to exchange $T_1$ and $L$.
>
> 2.  **Stage 2: Draft Correct & Final Verify (Parallel Execution)**
>     *   From the logits $L$, we extract the top-k tokens for each position (e.g., for $l_1$, we get $\{t_1^1, \dots, t_1^k\}$). This forms a candidate set $T_e$ with a logical dimension of $[|T_0|, k]$, which serves as a preliminary verification result for $T_0$.
>     *   The sequences $T_0$, $T_1$, and the flattened tokens from $T_e$ are concatenated to form $T_2$. As illustrated in Figure 3, this is logically organized as a multi-branch structure.
>     *   **Draft Correct**: This stage takes $T_2$ as input. Using an attention mask, it performs parallel inference to generate subsequent draft token sequences ($T_3$) for each branch. This is the **Tree Attention** mechanism shown in Figure 3 of our paper, a technique validated in prior work such as Medusa.
>     *   **Final Verify**: This stage takes the cached hidden states $F$ as input and runs them through the remaining layers of the target model to produce the final, authoritative verification result, $T_F = \{t'_1, t'_2, t'_3, t'_4\}$.
>
> 3.  **Final Verification and Path Selection**
>     *   The results are synchronized. The ground truth $T_F$ is used to validate the candidate paths formed by $\{T_0, T_e\}$. The correct branch is selected, as shown in Figure 3.
>     *   The subsequent draft tokens from the correct branch in $T_3$ will then be verified by the target model in the next forward pass. This ensures that the draft model's sequence to be verified always stays ahead of the target model's.

---

> ### Author Response · Authors · 2025-11-21
> **Response to Reviewer RFZF(part 2/3)**
>
> ### 2. What is the exact evaluation setup you used? Did all methods have access to the same computational resources?
>
> Thank you for asking. Our setup was consistent for all model combinations and evaluations. All experiments were conducted using **4 x H100 GPUs**, and we set **top-k=4** for methods that require it. All methods had access to the exact same computational resources to ensure a fair comparison. If there are any other specific setup details you would like to know, please do not hesitate to ask, and I will be happy to provide them.
>
> ***
>
> ### 3. Does FOLD actually end up using more GPU hours?
>
> Thank you for raising this important point. To clarify:
> (i) No, FOLD does not consume more GPU hours.
> (ii) Parallel execution does not necessarily require more GPUs.
>
> If by 'GPU hours' you mean the product of inference duration and the number of GPUs, then our experiments do not result in higher GPU hours. All our experiments were conducted on the same hardware setup (4 x H100 GPUs), which is identical to the evaluation setup used in the PEARL paper. Therefore, in terms of results, the total GPU hours are directly comparable.
>
> Regarding your question about whether parallel execution needs more GPUs, the multi-process parallelism is managed via the `accelerator`. In this setup, all allocated GPUs are visible and accessible to each process. Processes can efficiently share workloads on the same physical GPUs. This is supported by experiments in Appendix E of the PEARL paper, which discuss and validate the management of GPU resource contention in such setups.
>
> ***
>
> ### 4. About the *Weakness* section: The authors characterize FOLD as a “training-free method” (line 339), but FOLD requires the training of the Early Exit Module.
>
> I apologize if my wording caused a misunderstanding. The term "training-free method" mentioned in line 339 refers to the **baselines** we compared against, not FOLD itself.
>
> FOLD enhances the speculative decoding process by introducing a trainable Early Exit module, but it does not modify the original draft and target models. This module's primary role is to correct erroneous draft tokens, not to generate them. Under this condition, the ultimate speedup of any speculative sampling method is still fundamentally determined by the consistency between the draft and target models. FOLD's contribution is to raise the theoretical performance ceiling for a *given* target model being accelerated.
>
> Therefore, we concluded that comparing FOLD against "training-free" methods is fair because we are evaluating its ability to enhance a fixed model pairing. Furthermore, we did not shy away from comparing FOLD with methods that are not training-free. As shown in **Table 1**, we included the performance of [EAGLE2](https://arxiv.org/pdf/2406.16858) as a reference and demonstrated that with a suitable model combination (e.g., Llama3-1B & 70B), FOLD can achieve a superior speedup.

---

> ### Author Response · Authors · 2025-11-21
> **Response to Reviewer RFZF(part 3/3)**
>
> ### 5. Experiments are restricted to Llama 3 and Llama 2. Results on models from other model providers would strengthen the results by demonstrating generalizability.
>
> We agree that demonstrating generalizability is crucial. In response, we have conducted additional experiments on the Vicuna model family, the results of which are available in our detailed response to **Reviewer R4Ma**. Despite the questionable quality of the Vicuna-68m model(as our response to Reviewer R4Ma in section `1. FOLD Scalability to 13B Models`), our results show that FOLD still provides a noticeable acceleration.
>
> ***
>
> ### 6. No confidence intervals are provided for the speedup figures, making it difficult to contextualize the significance of the speedups.
>
> Thank you for this observation. As FOLD is a lossless speculative decoding acceleration method, we followed the convention established by nearly all prior work in this specific field, which typically does not report confidence intervals.
>
> The speedup ratio itself is considered the most critical and central metric in this domain. It accurately reflects the consistency between the draft and target models under various scenarios. I would like to add that our reported speedup ratios are the average of multiple runs, and they provide a direct and intuitive measure of a method's superiority.

---

> > ### Comment · Reviewer_RFZF · 2025-11-24
> >
> > Thank you for the detailed response.
> >
> > ### Clarifications [1-3]
> >
> > I found the clarification of the FOLD architecture provided in your response to be much clearer than the description currently in the paper, and would recommend it be included in future revisions as well. The other clarifications are also helpful and should similarly be included.
> >
> >
> > ### Training-free Baselines [4]
> >
> > The distinction drawn between the amount of training needed for the adapter in FOLD and methods like EAGLE could be made clearer. It would be good to baseline against methods that perform a comparable amount of training. For example, as you base the Early Exit module off of Kangaroo (Liu et al., 2024), that would be a good baseline. Further, I concur with Reviewer R4MA that comparisons with tree-based speculative decoding methods are necessary.
> >
> > ### Other Model Families [5]
> >
> > Thank you for your results on the Vicuna model family. I agree these results are evidence of FOLD's generalizability. If able, I encourage the authors to additionally evaluate on the Qwen 3 model family, as the wider availability of model sizes can help address the authors' concerns about the weakness of the draft model.
> >
> > ### Confidence Intervals [6]
> >
> > While related work tends not to include confidence intervals, their inclusion would make for a stronger paper. Given that you are already reporting averages over multiple runs, hopefully further reporting the number of runs and/or the variance over those runs will not be too difficult.
> >
> > Though the response addresses most of my concerns with the presentation of the work, given the scale of the revisions needed to integrate the clarifications, as well as my other concerns, I will keep my score.

---

### Official Review · Reviewer_R4Ma · 2025-11-01

**Soundness:** 3
**Presentation:** 3
**Contribution:** 3
**Rating:** 4
**Confidence:** 4

**Summary:**

This paper introduces FOLD, a novel method that enhances the speculative decoding process. It refines the two fundamental phases of speculative decoding—draft and verify—into a more granular four-step procedure: early draft, early verify, draft correct, and final verify. This refined, four-stage design is central to the method's effectiveness. It enables a subset of the tokens produced during the initial early draft phase to be supplemented and corrected using partial information from the target model, thereby substantially improving the quality of the draft tokens. Empirical results demonstrate the efficacy of the proposed method.

**Strengths:**

1. The central idea of this paper is both novel and sound, and the experimental results validate the effectiveness of the proposed methodology.
2. The paper is well-written, and the figures and tables are clear and easy to follow.

**Weaknesses:**

1. I observe that all experimental results were derived using the 70B large-scale model. I am curious whether the proposed FOLD method remains applicable and effective at smaller model scales, such as 7B or 14B.
2. The comparison between FOLD and other baseline methods (like vanilla SD) appears to be potentially unfair, as the latter seemingly did not utilize the tree draft mechanism while FOLD did. A portion of FOLD's observed speedup directly stems from the tree draft. To strengthen the persuasiveness of the results, I recommend the authors enable the draft tree mechanism for all compared methods and then re-evaluate the performance accordingly.
3. Regarding the results presented in Table 2, I suggest the authors include the speed and the corresponding speedup data for Auto Regression decoding, similar to what is shown in Table 1. This would offer a more complete analysis and comparison of the results.
4. The theoretical speedup of this work is contingent upon a meticulously designed parallel mechanism. However, the authors' open-sourced repository seems only to contain the debug code for FOLD, omitting the implementation of the crucial parallel acceleration component. I kindly request the authors to either provide or clearly indicate the implementation details and location of FOLD's core parallel mechanism within the existing open-source code.

**Questions:**

Please see the weakness.

---

> ### Author Response · Authors · 2025-11-20
>
> Dear Reviewer,
> We deeply value your expertise in identifying these nuanced aspects, which have prompted us to refine our explanations and experimental presentation. Below are detailed responses to your additional queries.
>
> ### 1. FOLD Scalability to 13B Models
> We appreciate your interest in larger-scale validation. We conducted experiments using **Vicuna-13B** (target) and **Vicuna-68M** (draft) with *top-k=4* on dual H100 GPUs. Results are summarized below:
>
>
> | Method | gsm8k  | Speedup | humaneval | Speedup |
> |--------|--------|---------|-----------|---------|
> | ar     | 50.60  | 1.00$\times$    | 50.60     | 1.00$\times$    |
> | sd     | 65.82  | 1.30$\times$    | 69.05     | 1.36$\times$    |
> | pearl  | 76.76  | 1.52$\times$    | 76.22     | 1.51$\times$    |
> | fold   | **78.17** | **1.54$\times$** | **78.77** | **1.56$\times$** |
>
> For models within the same series (e.g., Llama 3), manufacturers typically release a 7B model as the smallest variant. It is challenging to find suitable, smaller-scale than official 7B models to serve as draft models. Officially released smaller models, trained on similar data and with comparable methodologies, exhibit excellent consistency with their larger counterparts. This strong alignment leads to remarkable speedups in speculative sampling, as evidenced by the outstanding performance of the Llama3.2-1B and Llama3.1-70B pairing in Tables 1 and 2 of our paper.
>
> In contrast, smaller models trained by third parties often show suboptimal consistency due to differing training data. Their performance, not only in terms of alignment but also in practical accuracy, frequently falls short of expectations. Consequently, pairing a 7B target model with a third-party small model yields less than ideal results. Even if FOLD corrects the current incorrect draft tokens, the draft model's subsequent token predictions remain unreliable, necessitating frequent interventions from the early exit module. This is not an ideal scenario for showcasing the method's potential.
>
> I would like to emphasize that this performance limitation stems primarily from the unavailability of a suitable draft model, rather than from a lack of generalizability in FOLD itself. On the contrary, as demonstrated by Equation 11 in our paper, FOLD is designed to unlock the theoretical upper bound of acceleration performance in current speculative sampling methods. We believe that with more appropriately paired models, the benefits of FOLD would be even more pronounced.
>
> ---
>
> ### 2. Baseline Implementation of Tree Attention
> Because FOLD is very ingenious, I may need to explain to you why baselines do not and cannot use tree attention like FOLD does.
>
> Technically, FOLD employs an early exit module to obtain multiple potential outcomes in advance, which are then used to correct draft tokens. This design ensures that each draft token generates only k additional branches, without creating extra branches for the subsequent draft sequence. In contrast, the baseline methods lack a pre-verification mechanism. This makes it impossible to determine the optimal moment to generate top-k branches or to directly construct a complete k-ary tree. As the draft sequence grows, complete k-ary tree would lead to an exponential increase in the number of tokens. Such an approach would not only preclude a fair performance comparison but also significantly increase latency and memory consumption due to being compute-bound, ultimately degrading performance.
>
> Furthermore, I would like to add that, compared to other excellent and distinctive works, the combination of the early-exit module and tree attention in FOLD is both natural and elegant. Our work's distinctiveness, much like outstanding methods such as [Medusa](https://arxiv.org/abs/2401.10774) and [Eagle](https://arxiv.org/pdf/2503.01840), lies in the appropriate and effective application of tree attention. Therefore, to ensure a fair and meaningful comparison, we regretfully chose to maintain the original configurations of the baselines in our experiments.

---

> ### Author Response · Authors · 2025-11-20
>
> ### 3. Auto-Regression Speedup in Table 2
> While Auto-Regression is the benchmark for the target model's inference speed and the basis for our speedup ratio comparison (rather than being a speculative decoding method itself), including its metrics directly in Table 2 would have made the results more intuitive. Our initial reasoning was that since the Auto-Regression speed is a fixed constant, stating it in Table 1 would be sufficient. However, we overlooked how this might affect the reader's ability to make immediate comparisons.
>
> We sincerely appreciate you pointing this out. We will add this data to the table in the revised version of our paper.
>
> ---
>
> ### 4. Code Implementation Details
> The core inference logic is located in src/engine.py under the function parallel_debug. Our repository is a modified version based on the ParallelSpeculativeDecoding codebase (available at [smart-lty/ParallelSpeculativeDecoding](https://github.com/smart-lty/ParallelSpeculativeDecoding)). To maintain compatibility with the original repository's evaluation commands, we have integrated the evaluation code for FOLD into this structure.
>
> Furthermore, we undertook a significant effort to refactor the inference code for both the draft and target models to meet our specific requirements. These modifications can be found in src/earlyexit/draft.py and src/earlyexit/modeling_llama_v4.51_target.py, respectively. We understand that this amount of custom work can pose challenges for review, and we hope it has not caused undue difficulty in understanding our implementation.
>
> As a final point, we have thoroughly verified the integrity of the anonymous code repository. We can confirm that it runs correctly and can be used to replicate our evaluation results.

---

> > ### Comment · Reviewer_R4Ma · 2025-11-21
> > **Official Comment by Reviewer R4Ma**
> >
> > Thank you for your detailed response; it has addressed some of my initial concerns. However, a few key issues remain that I believe warrant further clarification or analysis.
> >
> > **1. FOLD Scalability to 13B Models**
> >
> > The pairing of **Llama-3.1-8B-Instruct** (target) and **Llama-3.2-1B-Instruct** (draft) appears to be a highly suitable combination for achieving effective acceleration, given the authors' discussion on model family consistency. Could the authors please provide the experimental results for FOLD using this specific model pair?
> >
> > **2. Baseline Implementation of Tree Attention**
> >
> > I believe there are existing techniques to prevent the draft tree from becoming an exponentially increasing k-ary tree. For example, one could pre-define the tree structure and populate it with tokens during the drafting process [1], or implement dynamic pruning of the draft tree [2].
> >
> > Essentially, my core point is as follows: FOLD's key contribution appears to be integrating the target model's early-exit outputs into the draft tree, resulting in more accurate draft tokens. Since using a draft tree itself can inherently accelerate generation, even if all draft tokens originate from the draft model, it is crucial to **decouple the separate contributions** to speedup provided by the *draft tree structure* and the *draft correction mechanism*. This distinction would better illuminate the unique benefits of FOLD.
> >
> > [1] EAGLE: Speculative Sampling Requires Rethinking Feature Uncertainty
> >
> > [2] EAGLE-3: Scaling up Inference Acceleration of Large Language Models via Training-Time Test

---

> ### Author Response · Authors · 2025-11-28
>
> ### 1. Regarding the Lack of Evaluation with Llama3-1B & 8B
>
> Thank you for raising this important point regarding the potential pairing of Llama-3.1-8B-Instruct (target) and Llama-3.2-1B-Instruct (draft).
>
> We did consider this model pairing early in our research and performed a preliminary analysis. However, we found that the inference speed difference between these two models is minimal—less than a 2x factor, which is not worth to correcting wrong draft tokens.
> Unlike Eagle's draft model which features only a single decoder layer, the Llama3.2-1B architecture incorporates 16 decoder layers. Despite their comparable parameter sizes, the Llama3.2-1B with its increased number of decoder layers demonstrates less impressive performance in inference speed.
> For this reason, we concluded that this pairing would not serve as a representative or informative test case for our method's capabilities and thus did not proceed with a full set of experiments.
>
> ### 2. Regarding the Application of Tree Attention to Baselines
>
> We explored experimental design possibilities but ultimately concluded that applying FOLD's tree structure to vanilla SD would not permit a fair comparative analysis to decouple the individual contributions of the draft tree structure and the draft correction mechanism.
> In FOLD, if the Early Exit module corrects a token, the verification of its subsequent tokens is deferred to the next cycle. If we were to apply the same tree structure to vanilla SD and allow it to verify these subsequent tokens within the current cycle, it would effectively grant the baseline an unfair advantage by giving it a shorter verification loop (with only one verification round). See: [branches of FOLD](https://i.postimg.cc/BQLRfrLd/tree-attn-di-1-ye-drawio.png)
> On the other hand, if omitting successor tokens with correction tokens, the behavior degenerates to standard target model rejection - indistinguishable from conventional autoregressive decoding. See:[branches of vanilla SD](https://i.postimg.cc/YqGTB5Lc/tree-attn-di-2-ye-drawio.png)
>
> As a practical alternative to address this decoupling concern, **we have included an ablation study in the paper where FOLD operates in a `top-1` setting**. This configuration minimizes the influence of the tree attention mechanism, thereby better isolating and highlighting the performance gains attributable primarily to our draft correction mechanism.
>
> Furthermore, we observed that EAGLE compared the tree attention mechanism with chain-based approaches to evaluate the performance gain from tree attention. Such ablation studies are feasible for EAGLE due to two key characteristics:
> 1. All draft branches are generated by the draft model and verified within a single round, maintaining timing alignment comparable to other speculative sampling methods.
> 2. Building upon (1), the chain mechanism can be viewed as a top1 special case of tree attention, while tree attention allows flexible dynamic/static adjustment of topk configurations or tree topologies.
> In contrast, FOLD’s early exit module for draft token correction introduces fundamentally distinct constraints:
> 1. Early exit branches undergo delayed validation (next round) temporally.
> 2. Early exit branches represent static multi-branch structures requiring integration of preliminarily verified tokens with draft tokens - structurally necessitating tree attention for subsequent sequence generation.

---

### Official Review · Reviewer_qNND · 2025-11-01

**Soundness:** 3
**Presentation:** 3
**Contribution:** 2
**Rating:** 4
**Confidence:** 4

**Summary:**

The paper presents FOLD (Fast cOrrect specuLative Decoding), a novel framework that improves the efficiency of speculative decoding in Large Language Models by mitigating the high cost of token rejection. Unlike traditional methods that discard all computation after a rejected token, FOLD introduces a four-stage process—Early Draft, Early Verify, Draft Correct, and Final Verify—that proactively generates and verifies multiple alternative sequences using an Early Exit Module and Tree Attention. This allows the system to recover from errors without losing progress, turning rejection into correction. Compatible with existing approaches like Pearl, FOLD achieves up to a 4.09× speedup over standard auto-regressive decoding, marking a significant advancement in inference performance.

**Strengths:**

- Reframes speculative decoding by treating token rejection as an opportunity for rapid correction instead of failure to improve efficiency.
- Introduces a four-stage framework with an Early Exit Module and Tree Attention, enabling parallel correction and preserving useful computation.
- Demonstrates up to a 4.09× speedup over standard auto-regressive decoding and is easily adaptable to existing methods like Pearl, proving both performance and practicality.

**Weaknesses:**

- Some methodological details, such as hyperparameter choices, are insufficiently explained, making the approach harder to reproduce or tune.
- The proposed Tree Attention mechanism, while innovative, may introduce additional memory overhead during inference, potentially limiting scalability on resource-constrained hardware.
- The experiments focus on short-context datasets, leaving uncertainty about FOLD’s effectiveness and efficiency on longer or more complex tasks, which limits the demonstrated generalizability of the approach.

**Questions:**

- On line 186, there is “γ ×K + 1 draft branches”, would that be “γ ×k + 1”? Can you explain more on this?
- In the tree attention part, multiple branches will be generated, which increases the GPU memory usage/accesses at inference time. Considering the decoding stage in LLM inference, which is already memory-bound, will this tree attention further induce bottlenecks on limited bandwidth? Can you explain qualitatively or quantitatively?
- GSM8K and HumanEval are used in the experiment section. Does FOLD also work on datasets with longer prompt lengths, such as arxivSummary or other similar datasets, to further demonstrate the effectiveness of FOLD under different workloads?
- How does FOLD decide what values to be assigned for γ1 and γ2 at inference time? (i.e, does FOLD use fixed values for both across different prompts or does FOLD dynamically change values) For example, given two prompts of different difficulty levels (one is easy to answer/generate results, while the other is not), I assume the draft model in the Early Draft phase may be expected to generate different numbers of tokens (different γ1). Can you elaborate more on this point qualitatively or quantitatively?


Minor typos:
Line 88-89, farward -> forward
Line 459, parameterkindirectly -> parameter k indirectly
Line 461, differentkvalues -> different k values
Line 461, parameterkincreases -> parameter k increases
Line 464, increasingkto -> increasing k to

---

> ### Author Response · Authors · 2025-11-20
>
> Dear Reviewer,
>
> Thank you for your thoughtful review and valuable suggestions. We sincerely appreciate the time you dedicated to evaluating our work, and we have carefully addressed all your concerns below. Your insights have significantly helped us improve the clarity and completeness of our methodology.
>
> ### 1. Clarification on "γ × K + 1" (Line 186)
> To explain the "+1" term: During verification of a draft sequence of length *γ*, our Early Exit module generates logits across *γ* positions. For each position, we select the top-*k* probable candidates to create **γ × k** correction branches. Crucially, we retain the original draft branch since it may still be valid. Thus, the total branches become **γ × k + 1**, combining correction paths with the original candidate.
>
> ---
>
> ### 2. Memory Overhead Analysis of Tree Attention
> We appreciate your insightful question regarding memory bottlenecks. Our analysis confirms that Tree Attention **does not significantly increase memory-bound constraints**, as the additional overhead is marginal compared to baseline decoding costs. Below is a quantitative breakdown for **Llama2-7B**:
>
> | Component | Size Calculation | Memory Usage |
> |-----------------------------|------------------------------------------|-------------|
> | **Model Weights** (per token) | 7B params × 2 bytes | **14 GB** |
> | **KV Cache** (per token) | 32 layers × (4096 Key + 4096 Value) × 2 bytes | **0.524 MB** |
>
> The 14 GB weight transfer dominates memory bandwidth during token generation. While Tree Attention introduces extra tokens, their aggregated KV cache remains negligible (e.g., 100 tokens ≈ 52.4 MB) against the 14 GB baseline. Once a verification completed, only the KV cache of accepted tokens will be retained.
>
> ---
>
> ### 3. Evaluation on Long-Context Datasets
> We conducted additional experiments on **SpecBench**'s **summarization** and **RAG** tasks (long-prompt scenarios). Results are summarized below:
>
> | Model          | Method | Summary | Speedup | RAG | Speedup |
> |----------------|--------|---------------|---------|---------------|---------|
> | **Llama3-8B&70B** | AR     | 14.85         | 1$\times$      | 14.85         | 1$\times$     |
> |                | Pearl  | 42.16         | 2.84$\times$   | **40.79**         | **2.75$\times$**   |
> |                | FOLD   | **43.53**         | **2.93$\times$**   | 39.13         | 2.64$\times$   |
> | **Llama3-1B&70B** | AR     | 14.85         | 1×      | 14.85         | 1$\times$      |
> |                | Pearl  | 45.42         | 3.06$\times$   | **46.34**         | **3.12$\times$**  |
> |                | FOLD   | **50.44**         | **3.40$\times$**   | 46.17         | 3.11$\times$   |
> | **Llama2-7B&70B** | AR     | 13.77         | 1$\times$      | 13.77         | 1$\times$      |
> |                | Pearl  | 34.13         | 2.48$\times$   | 34.43         | 2.50$\times$   |
> |                | FOLD   | **36.66**         | **2.66$\times$**   | **36.27**        | **2.63$\times$**  |
>
>
> FOLD maintains competitive performance but may exhibit minor regressions in some RAG tasks with Llama3 series (e.g., -1.66% in RAG for Llama3-8B). This aligns with our discussion in Line 429: the current Early Exit module uses a simple design with simple training data for feasibility demonstration, not long-context optimization. In addition, there is a specific research direction focused on [*long-context-speculative-decoding*](https://github.com/hemingkx/SpeculativeDecodingPapers?tab=readme-ov-file#long-context-speculative-decoding). Objectively speaking, we do not belong to this direction. Thank you for your understanding.
>
> ---
>
> ### 4. Hyperparameter Settings for γ₁ and γ₂
> We thank you for this nuanced question. In our framework, **γ₁ and γ₂ are static values** determined by two factors:
> 1. Target/Draft model speed ratio.
> 2. Proportion of First *N* Layers (input to Early Exit).
>
> For optimal balance between Early Exit accuracy and draft token volume:
> $$\gamma_1 = \gamma_2 = \frac{\gamma}{2} \quad \text{(e.g., } \gamma=4 \Rightarrow \gamma_1=\gamma_2=2\text{)}$$
> If the speed ratio is odd, we adjust *N* to approach (but not exceed) $\frac{Total Layers}{2}$.
>
> Regarding dynamic γ-values for varying prompt difficulties: we think it is an interesting direction. However, reliably quantifying complexity and difficulty of prefix sequence remains an open challenge, which is required by achieving dynamic $\gamma_1$ and $\gamma_2$. If it is posiible, there will be lots of method to improve our framework like multi-draft structure.
>
> ---
>
> We hope these explanations fully address your concerns. Thank you again for your invaluable feedback, which has strengthened our work. We look forward to incorporating these clarifications in the final manuscript.
>
> Best regards, author.

---

### Note · Authors · 2026-01-27

I have read and agree with the venue's withdrawal policy on behalf of myself and my co-authors.

---

### Meta-Review · Area_Chair_EVZE · 2025-12-27

**Summary:**

This paper proposes a new method called FOLD (Fast cOrrect specuLative Decoding) that utilizes a four stage process to proactively generates and verifies multiple alternative sequences using an Early Exit Module and Tree Attention. Compared with the default approach, this method does not waste all unselected tokens. The reviewers' main concerns are on the experiment settings. E.g., Some methodological details are missing and the experiments are on a relatively strong scale. Hence, I agree that this paper should be further improved to meet the bar.

**Reviewer Concerns:**

1. Not comprehensive enough experiments
2. The proposed method might introduce additional memory usage issues
3. The experiment scale is not big enough.

**Reviewer Scores:**

I have carefully read all rebuttals and reviewer opinions and do not see strong signals indicating a change in scores.

---

### Decision · Program_Chairs · 2026-01-26

Reject